# Endothelial Cell Dysfunction and Hypoxia as Potential Mediators of Pain in Fabry Disease: A Human-Murine Translational Approach

**DOI:** 10.3390/ijms242015422

**Published:** 2023-10-21

**Authors:** Katharina Klug, Marlene Spitzel, Clara Hans, Alexandra Klein, Nicole Michelle Schottmann, Christoph Erbacher, Nurcan Üçeyler

**Affiliations:** 1Department of Neurology, University Hospital of Würzburg, Josef-Schneider-Str. 11, 97080 Würzburg, Germany; klug_k1@ukw.de (K.K.); spitzel_m@ukw.de (M.S.); clara.hans@stud-mail.uni.wuerzburg.de (C.H.); schottmann_n@ukw.de (N.M.S.); erbacher_c@ukw.de (C.E.); 2Würzburg Fabry Center for Interdisciplinary Therapy (FAZIT), University Hospital of Würzburg, 97080 Würzburg, Germany

**Keywords:** iPSC, hypoxia, vasculopathy, mitochondriopathy, Fabry disease, endothelial cells, *GLA* KO mouse model, blood vessels, dorsal root ganglion

## Abstract

Fabry disease (FD) is caused by α-galactosidase A (AGAL) enzyme deficiency, leading to globotriaosylceramide accumulation (Gb3) in several cell types. Pain is one of the pathophysiologically incompletely understood symptoms in FD patients. Previous data suggest an involvement of hypoxia and mitochondriopathy in FD pain development at dorsal root ganglion (DRG) level. Using immunofluorescence and quantitative real-time polymerase chain reaction (qRT PCR), we investigated patient-derived endothelial cells (EC) and DRG tissue of the *GLA* knockout (KO) mouse model of FD. We address the question of whether hypoxia and mitochondriopathy contribute to FD pain pathophysiology. In EC of FD patients (P1 with pain and, P2 without pain), we found dysregulated protein expression of hypoxia-inducible factors (HIF) 1a and HIF2 compared to the control EC (*p* < 0.01). The protein expression of the HIF downstream target vascular endothelial growth factor A (VEGFA, *p* < 0.01) was reduced and tube formation was hampered in the P1 EC compared to the healthy EC (*p* < 0.05). Tube formation ability was rescued by applying transforming growth factor beta (TGFβ) inhibitor SB-431542. Additionally, we found dysregulated mitochondrial fusion/fission characteristics in the P1 and P2 EC (*p* < 0.01) and depolarized mitochondrial membrane potential in P2 compared to control EC (*p* < 0.05). Complementary to human data, we found upregulated hypoxia-associated genes in the DRG of old *GLA* KO mice compared to WT DRG (*p* < 0.01). At protein level, nuclear HIF1a was higher in the DRG neurons of old *GLA* KO mice compared to WT mice (*p* < 0.01). Further, the HIF1a downstream target CA9 was upregulated in the DRG of old *GLA* KO mice compared to WT DRG (*p* < 0.01). Similar to human EC, we found a reduction in the vascular characteristics in *GLA* KO DRG compared to WT (*p* < 0.05). We demonstrate increased hypoxia, impaired vascular properties, and mitochondrial dysfunction in human FD EC and complementarily at the *GLA* KO mouse DRG level. Our data support the hypothesis that hypoxia and mitochondriopathy in FD EC and *GLA* KO DRG may contribute to FD pain development.

## 1. Introduction

Fabry disease (FD) is a rare lysosomal storage disorder caused by several hundred variants in the α-galactosidase A gene (*GLA*) coding for the α-galactosidase A enzyme (AGAL) [1]. AGAL is involved in the cleavage of glycosphingolipids such as globotriaosylceramide (Gb3). Due to AGAL impairment in FD, Gb3 accumulates in patients’ lysosomes, which results in a multiorgan disorder including cardiomyopathy, nephropathy, and cerebral stroke. The peripheral nervous system is involved in terms of small fiber neuropathy, which may lead to triggerable acral burning pain [2]; however, the exact mechanisms underlying FD pain are incompletely understood.

There is evidence for impaired endothelial cell (EC) function and integrity in FD. Several in vitro studies on FD-patient-derived EC demonstrated lysosomal Gb3 accumulation and aberrant angiogenesis [3,4,5]. Studies investigating *GLA* knockout (KO) mice [6] revealed similar data [7,8]. In FD patients, malperfusion of the dorsal root ganglia (DRG) was found using magnetic resonance imaging (MRI) [9], potentially indicating hypoxia at the DRG level. Also, the impact of mitochondrial dysfunction on FD was reported in animal and human studies [10,11,12], which might develop subsequently due to the hypoxic environment at DRG level. However, the mechanisms that contribute to the interaction between vasculopathy, hypoxia, mitochondrial dysfunction, and the eventual development of FD pain are not yet fully understood.

We assume that dysregulated respiratory pathways, including mitochondrial dysfunction, in FD [11,12] may impair cellular metabolism and thereby induce a hypoxic environment at DRG organ level [13,14]. We hypothesize that FD vasculopathy is influenced by an interplay of mechanisms causing aberrant angiogenesis and consecutive tissue-specific hypoperfusion contributing to FD pain. To test this hypothesis, we investigated EC function, morphology, and hypoxia-associated mediators in an FD human-murine translational approach.

## 2. Results

### 2.1. Investigations in Patient-Derived EC

#### 2.1.1. Clinical Characteristics

Table 1 summarizes the main clinical characteristics of our study subjects. We investigated two male patients carrying a non-sense variant in the *GLA* gene (P1 and P2) and a healthy male control (Ctrl) without genetic variants in *GLA*. P1 showed a severe FD phenotype with nephropathy, cardiomyopathy, and FD-associated pain already occurring in childhood. Thermal perception thresholds were elevated in QST and the distal IENFD was reduced to 4.9 fibers/mm (laboratory reference value of the lower leg: 9 ± 3 fibers/mm). P2 did not show any organ involvement, reported no FD pain, and had normal thermal perception thresholds in QST. The IENFD at the lower leg was reduced to 3.7 fibers/mm.

#### 2.1.2. FD-Patient-Derived EC Carry a Characteristic Cellular Phenotype

iPSC lines were generated from skin fibroblasts of all three study participants as previously described [15] and successfully differentiated into EC, which showed expression of CD144 and CD31 as specific EC markers (Figure 1A–C). LDL uptake was sufficient in all cell lines (Figure 1D). Gb3 accumulated and AGAL activity was reduced in the EC of both FD patients P1 (*p* < 0.01) and P2 (*p* < 0.05) (Figure 1E,F) compared to the healthy control. Exemplified expansion microscopy with CD144 and CD31 revealed functional endothelial junctions in EC of the healthy control and P1 (Figure 1G).

#### 2.1.3. Hypoxia-Associated Gene Expression Is Modified in P1 Compared to Control EC

We compared relative gene expression of the selected hypoxia-associated genes in P1’s and the control’s EC. Relative quantification revealed 1.5-fold higher gene expression of DNA damage-inducible transcript 4-like (*DDIT4L*) and its paralog *DDIT4*, angiopoietin-like 4 (*ANGPTL4*), insulin-like growth factor binding protein 1 (*IGFBP1*), and adrenomedullin (*ADM*) in the EC of P1 compared to the control’s EC (Figure 2A). Gene expression of vascular endothelial growth factor A (*VEGFA*) was reduced 3.5 times over in P1’s EC compared to the control EC (Figure 2A). Validation of selected target genes, including hypoxia-sensing factors *HIF1a* and *HIF2* and their downstream targets *ADM* and *VEGFA,* using a single PCR test showed no difference between P1’s, P2’s, and the control EC (Figure 2B).

#### 2.1.4. Reduced Nuclear HIF1a and Increased HIF2 Signals in EC of P1 and P2 Compared to Control EC

P1, P2, and control EC showed higher nuclear translocation of the immunofluorescent signal of HIF1a after cultivation under hypoxia compared to all groups under normoxic conditions (Figure 3A, *p* < 0.001). However, the translocation ratio of HIF1a in the EC of P2 was reduced compared to the control EC and P1 EC (Figure 3A, control: *p* < 0.001, P1: *p* < 0.05). The nuclear HIF2 signal was higher in the EC of P1 and P2 compared to control EC (Figure 3B, P1: *p* < 0.01, P2: *p* < 0.001). Complementarily, VEGFA protein expression in the P1 and P2 EC was reduced compared to in the control EC (Figure 3C, P1: *p* < 0.01, P2: *p* < 0.001).

#### 2.1.5. Impaired Tube Formation Properties in FD-Patient-Derived EC under Normal and Hypoxic Conditions

We then investigated the physiological tube formation properties of human EC cultures. P1-derived EC showed a reduction in the total tube length and branching of tubes compared to control EC (*p* < 0.01 each), while no such differences were found when comparing the P2-derived EC to control cells or to P1-derived EC (Figure 4A,B). Incubation with agalsidase-α did not change the cellular phenotype (Figure 4A,B).

#### 2.1.6. Selective TGFβ Pathway Block Recovers FD-Patient-Derived EC Function

We performed experiments targeting the transforming growth factor β-(TGFβ) pathway as a general regulator of vascularization. Incubation with the selective activin receptor-like kinase 5 (ALK5) inhibitor SB, which is part of the TGFβ pathway, restored the cellular phenotype in the P1-derived EC compared to the control conditions (Figure 5A,B).

#### 2.1.7. Selective TGFβ Pathway Inhibition Revealed Differential Expression of ALK5 Downstream Target Genes

Further, gene expression analysis of ALK5-dependent TGFβ pathway downstream targets endoglin (*ENG*), caveolin-1 (*CAV*), leucine-rich alpha-2-glycoprotein 1 (*LRG1*), and *TGFβ1* was performed and revealed a reduction in the caveolin-1 expression in the EC of P1 compared to the EC of the healthy control (Figure 5C, *p* < 0.05).

#### 2.1.8. Mitochondrial Membrane Potential and Form Are Altered in FD Compared to Control EC

We investigated the mitochondrial morphology and membrane potential in the EC from both patients compared to the mitochondria from the healthy control EC. For this, we immunolabeled the mitochondria against TOMM20 (Figure 6A) and performed a JC10 assay to determine the mitochondrial membrane potential (Figure 6B). The reduction, i.e., depolarization, of the mitochondrial membrane potential (*p* < 0.05) in EC of P2 compared to EC from P1 and the healthy control points to more bioenergetic stress in the EC of P2. Using the MitoHacker software [16], we found more roundish mitochondria in EC of P1 compared to control EC (*p* < 0.001; Figure 6C). In P2, mitochondria were more elongated and branched compared to control EC (*p* < 0.001). In addition, we found a higher value for solidity in P1 compared to P2 and the healthy control EC (*p* < 0.001). In both patients’ EC, but even more so in P2’s, mitochondria lay more distant from the cellular nucleus than in EC from the healthy control (P1: *p* <0.01, P2: *p* <0.001).

### 2.2. Investigations in Murine DRG Tissue and Neuronal Cell Culture

#### 2.2.1. Higher Expression Levels of the Hypoxia-Associated Target Genes in the DRG of Old GLA KO Than Old WT Mice

We compared DRG gene expression in old *GLA* KO and WT mice of the hypoxia-associated targets (Figure 7) and found five hypoxia-associated target genes upregulated in the DRG of old *GLA* KO mice compared to old WT mice: *DDIT4* (Figure 7B, *p* < 0.001), FBJ osteosarcoma oncogene (*FOS*, Figure 7C, *p* < 0.001), hexokinase 2, (*HK2*, Figure 7D, *p* < 0.05), lectin, galactose binding, soluble 3, (*LGALS3*, Figure 7E, *p* < 0.001), and transferrin receptor (*TFRC*, Figure 7F, *p* < 0.01).

#### 2.2.2. Increased Nuclear HIF1a Intensity in Old *GLA* KO DRG Compared to Old WT DRG

Next, we investigated whether HIF1a was disturbed at protein level in *GLA* KO DRG neurons (Figure 8). HIF1a^+^ neurons were quantified per mm^2^ cell-body-rich area (CBRA) of the DRG (Figure 8A–D). In young and old *GLA* KO DRG, the number of cytosolic HIF1a^+^ neurons was lower compared to young (Figure 8E, *p* < 0.05) and old WT (Figure 8F, *p* < 0.01) DRG, respectively. Further, we assessed the nuclear intensity of HIF1a in the DRG neurons of young and old *GLA* KO and WT mice (Figure 9A,B). The nuclear HIF1a intensity did not differ between DRG neurons of young WT and young *GLA* KO mice (Figure 9C, *p* > 0.05), while in DRG neurons of old *GLA* KO mice compared to old WT mice, nuclear HIF1a intensity was higher (Figure 9D, *p* < 0.01).

#### 2.2.3. Higher Numbers of CA9^+^ DRG Neurons in Old *GLA* KO Mice Compared to Old WT Mice

We assessed the protein expression levels of the HIF1a downstream target carbonic anhydrase 9 (CA9) in murine DRG. We quantified the number of CA9^+^ cells per mm^2^ CBRA in young and old *GLA* KO and WT mice (Figure 10A,B). The number of CA9^+^ neurons did not differ between the DRG of young WT and young *GLA* KO mice (Figure 10C), while the number of CA9^+^ neurons in the DRG of old *GLA* KO mice was higher compared to old WT mice (Figure 10D, *p* < 0.01).

#### 2.2.4. Higher Expression Levels of Hypoxia-Associated Target Genes in Cultivated DRG Neurons of WT and *GLA* KO Mice under Hypoxia Compared to Normoxia

We analyzed gene expression levels of the hypoxia-associated target genes in the DRG neuronal cell cultures of old WT and *GLA* KO mice cultivated for 24 h under normoxia and hypoxia. We found five hypoxia-associated target genes upregulated in the DRG neurons of WT and *GLA* KO mice under hypoxia compared to those under normoxia (Figure 10). For *DDIT4*, *HK2*, and *LDHA*, no intergroup difference was found (Figure 11A–C), while *TFRC* gene expression was upregulated only in the WT DRG neurons under hypoxia (Figure 11D). *LGALS3* gene expression was upregulated in the *GLA* KO DRG neurons compared to the WT DRG neurons under normoxia and displayed higher gene expression levels under hypoxia (Figure 11E).

#### 2.2.5. Similar HIF1a Protein Distribution in DRG Neurons of Old WT and *GLA* KO Mice Cultivated under Normoxia and Hypoxia

HIF1a protein distribution was additionally assessed in the DRG neuronal cell culture of old WT and *GLA* KO mice under normoxic and hypoxic conditions (Figure 12A–D). We found no differences in the amount of HIF1a^+^ (Figure 12E) or in the HIF1a nuclear intensity (Figure 12F) between genotypes and experimental conditions.

#### 2.2.6. Similar Number of CA9^+^ DRG Neurons in WT and *GLA* KO Mice Cultivated under Normoxia and Hypoxia

Complementary to the CA9^+^ neuronal count in DRG tissue, we assessed the number of CA9^+^ neurons in old WT and *GLA* KO mice cultivated for 24 h under normoxia and hypoxia (Figure 13A–D). We found no differences with respect to the number of positively counted neurons between genotypes or experimental conditions (Figure 13E).

#### 2.2.7. Reduced Vascularization in Old *GLA* KO Compared to Old WT Mice

We quantified the DRG cross-sectional CBRA covered with blood vessels in young and old *GLA* KO and WT mice (Figure 14A–D). While no intergroup difference was found in young mice, the CBRA in DRG of old *GLA* KO mice was greater compared to old WT mice and young GLA KO mice (Figure 14E, both *p* < 0.05). Additionally, the total blood vessel area covering the CBRA in DRG of old *GLA* KO mice was smaller compared to that of old WT mice (Figure 14F, *p* < 0.05). Analysis of the number of branches revealed fewer blood vessel branches in the DRG of old *GLA* KO mice compared to DRG of old WT mice (Figure 14G, *p* < 0.05). The total blood vessel length was also reduced in DRG of the old *GLA* KO mice compared to those of young *GLA* KO mice (*p* < 0.01) and old WT mice (*p* < 0.001) (Figure 14H). Measurement of the blood vessel thickness in the DRG of the young and old *GLA* KO and WT mice showed no differences (Figure 14I).

## 3. Discussion

We have investigated potential mechanisms linking pain in FD with hypoxia based on vasculopathy at the DRG level. We provide evidence for impaired angiogenesis and EC integrity, which may reciprocally affect intracellular hypoxia in EC and DRG neurons.

Hypoxia is known as the main driver of angiogenesis [17,18], with ample evidence that induced hypoxic mechanisms and aberrant angiogenesis may particularly contribute to FD pathophysiology [3,4,5,19,20]. However, no direct evidence for the involvement of hypoxia in FD pain development is available. Our data show differential gene expression in human FD EC of target genes which are known to contribute to autophagy, cellular stress [21,22], impaired lipid metabolism [23,24], cellular trafficking [25], altered angiogenesis, and EC integrity [26,27,28,29]. Complementary protein data on control and FD EC support gene expression data on the activated hypoxic mechanisms reflected by decreased HIF1a and increased HIF2 in FD EC. While HIF1a protein expression is mainly elevated in acute hypoxia, it is downregulated in chronic hypoxia, followed by HIF2 protein upregulation [30,31,32]. These results point to the involvement of chronic hypoxia in human FD EC. Dysregulation of these genes and protein targets can further contribute to cellular stress and DNA damage. DNA damage-associated targets have already been investigated in human FD studies [33,34] reporting that lyso-Gb3, a deacylated form of Gb3, induces cellular stress, leading to FD human kidney cell DNA damage. FD patients also display neurovascular involvement such as cerebral stroke and microangiopathy [35,36,37]. Thus, we investigated different EC properties in control EC and EC derived from FD patients reporting either pain attacks (P1) or no pain (P2). In both sets of FD-patient-derived EC, we demonstrated that EC characteristics were disturbed, including VEGFA protein expression, tube formation, and total branch length. Other in vitro studies reported similar findings on EC disturbance in FD patients [3,4,5,38], while serum of FD patients [39,40,41,42] and kidney tissue of FD mouse models [3,43] revealed an increase in VEGFA protein levels. We suggest that decreased VEGFA protein levels in P1’s and P2’s EC cultures might have occurred due to the involvement of chronic hypoxia, reflected by elevated HIF2 protein levels [30,31]. Further work investigated the role of the ALK5-dependent TGFβ pathway as an additional mechanism in angiogenesis [3,44]. This pathway can be modulated via the ALK5-inhibitor SB. We performed inhibition experiments with the control and FD EC and demonstrated restoration of the tube formation ability of FD EC, while treatment with agalsidase-α was not effective. Additionally, *CAV1*, a key player in the ALK5-dependent TGFβ pathway, was downregulated in P1’s EC. *CAV1* is the main driver of caveolae biogenesis, which is mainly located at cellular lipid rafts [45], pointing to the involvement of this specific molecule and pathway in altered FD EC morphology [46]. Another study reported that CAV1 protein levels can be upregulated by adding acetylsalicylic acid as a supporting treatment to chaperone therapy for FD-patient-derived fibroblasts [47], revealing *CAV1* as an interesting target in FD pathophysiology.

Another mechanism, which could be a potential metabolic contributor to vascularization and EC dysfunction in vivo, is mitochondrial dysfunction as reported in FD patients and the FD zebrafish model [10,11,12,48]. Mitochondrial size and form may indicate the balance between fission and fusion in the cells [49,50,51]. We found that mitochondria in the EC of P1 were smaller and had fewer branches compared to mitochondria in the EC of the healthy control. This might indicate a shift toward fission, an additional factor for stressed and apoptotic cells [52]. EC of P2 showed more branched and bigger mitochondria than EC of P1. Both mitochondrial states lead to an imbalance in the orchestration of fission and fusion, causing a prolongation of the cell cycle and impairing cellular metabolism [49,50,51,53].

Complementary to the human EC data, we report findings in the *GLA* KO mouse model investigating hypoxic and angiogenic disturbances at the gene, protein, and vascularization levels in DRG. Gene expression data revealed the upregulation of the hypoxia-associated target genes in whole DRG samples and DRG neuronal cell cultures associated with autophagy [54,55], cellular stress [56], neuronal activity, hypoxia [57,58], and lysosomal membrane damage [59] similar to FD EC. In previous work, increased apoptosis levels and cellular stress were found in the DRG neurons of *GLA* KO mice compared to WT mice [60], supporting our results of dysregulated DNA-damage- and cellular-stress-associated genes in human FD EC.

Analysis of the protein expression in *GLA* KO DRG neurons revealed increased HIF1a nuclear translocation and higher numbers of CA9^+^ neurons only in whole DRG tissue, which hints at the involvement of chronic hypoxia rather than acute hypoxia under in vivo conditions in the pathophysiology of the *GLA* KO mouse model [30,32,61]. It is suggested that a hypoxic environment can influence the functionality and expression of different ion channels involved in the development of pain [2,62,63,64,65]. As previous studies have demonstrated, impairment of the functionality and expression of ion channels like transient receptor potential vanilloid 1 (TRPV1), transient receptor potential ankyrin 1 (TRPA1), calcium-activated potassium channel 3.1 (KCa3.1), and voltage-gated sodium channel (NaV) 1.7 and 1.8 is associated with FD pain-like development in especially the *GLA* KO mouse model [60,66,67,68]. Those studies, and our previously and currently generated data, support the hypothesis that active hypoxic mechanisms might play a crucial role in the induction and maintenance of an FD pain-like phenotype of the *GLA* KO mouse model. Further, under chronic hypoxia, HIF1a translocates into the nucleus and acts there as a transcription factor upregulating hypoxia-associated genes and CA9 [32], which is a downstream target of HIF, is known to be a cellular hypoxia sensor [69,70], and is associated with neuropathic pain [57]. In contrast, under acute hypoxia, HIF1a is still located within the cytosol [32], suggesting that its influence on the protein expression modification of downstream targets was yet not visible in HIF1a distribution and CA9 regulation after 24 h of hypoxia in our study on the *GLA* KO mouse DRG neurons. Other studies report HIF-independent mechanisms, e.g., the involvement of miRNAs, NFκB, or PI3K/Rho/Rho kinase, influencing the gene expression levels of the hypoxia-associated target genes. Especially Rho kinase is associated with angiogenesis [71]. Similar to human EC, *GLA* KO mice displayed impaired vascularization at the DRG level as represented by lower blood vessel branching and length. This vasculopathy might result from HIF-dependent or HIF-independent mechanisms, e.g., the involvement of cyclooxygenase-2 (COX2) in the modulation of angiogenesis [71].

Our data on *GLA* KO mice at the DRG level reflect a potential involvement of hypoxia and vascular impairment in FD EC. Vasculopathy, potentially resulting from activated hypoxic mechanisms, can be linked to ion channel alterations [72] and pain development [73]. Another study reported TGFβ-dependent alleviation of vascular endothelial dysfunction via the application of the Rho kinase inhibitor Fasudil in vivo and in vitro and the improvement of several FD symptoms in a modified *GLA* KO mouse model [38]. We assume that a hypoxic environment contributes to hypoperfusion at the DRG organ level via altered angiogenesis, which might contribute to FD pain development.

## 4. Materials and Methods

### 4.1. Experiments on Human Biomaterial

#### 4.1.1. Patients and Clinical Examination

For the human studies, the biomaterial of two male patients with genetically confirmed FD (P1 and P2) with and without FD pain and a healthy male control (Ctrl) with normal *GLA* genetics were investigated as previously characterized [15]. In brief, the patients were recruited at the Fabry Center for Interdisciplinary Therapy (FAZIT), University of Würzburg. All patients underwent neurological examination and filled in the Würzburg Fabry Pain Questionnaire (FPQ) [74]. Nerve conduction studies of the sural nerve were performed to exclude polyneuropathy and quantitative sensory testing (QST) was used to determine the individual sensory profile. A 6 mm skin punch biopsy was obtained from the lateral lower leg. One half of the biopsy was used to determine the diagnostic intraepidermal nerve fiber density (IENFD), and the other half served to cultivate fibroblasts as previously described [75]. For this, the dermal and epidermal skin layers were mechanically separated, and the dermis was cultivated in a fibroblast culture medium (DMEM/F12 + 100 U/mL of penicillin/100 µg/mL of streptomycin [PenStrep; both Thermo Fisher Scientific, Waltham, MA, USA] + 10% fetal calf serum [FCS; Merck, Darmstadt, Germany]). Our study was approved by the Würzburg Medical Faculty Ethics Committee (#135/15) and subjects were enrolled after written informed consent.

#### 4.1.2. Generation of Subject-Derived EC

Induced pluripotent stem cells (iPSC) were generated from the dermal fibroblasts of all three subjects and were further differentiated into EC based on a modified protocol [76]. The iPSC were seeded on extracellular matrix-coated plates (Corning, Corning, NY, USA) in an iPSC cultivation medium (StemMACS™ iPS-Brew XF [Miltenyi Biotec, Bergisch Gladbach, Germany], 1% PenStrep supplemented with 10 μM of Y27632 [Miltenyi Biotec, Bergisch Gladbach, Germany]) and cultivated until 75% confluence was reached. The mesodermal germ layer was induced using a mesoderm induction medium (RPMI 1640 medium [Thermo Fisher Scientific, Waltham, MA, USA], B-27 supplement minus insulin [Thermo Fisher Scientific, Waltham, MA, USA], and CHIR99021 [Axon Medchem, Groningen, NL]) for four days, with a stepwise reduction in CHIR by 50% to 3 µM. From day five to eight, the EC were differentiated (EGM 2 BulletKit [Lonza, Basel, Switzerland], human VEGF-165 [Peptrotech, Rocky Hill, NJ, USA], human FGF-basic 154 [Peprotech, Rocky Hill, NJ, USA], and SB-431542 [Miltenyi Biotech, Bergisch Gladbach, Germany]) and purified using magnetic activated cell sorting (MACS, Miltenyi Biotech, Bergisch Gladbach, Germany) with CD144 magnetic beads (Miltenyi Biotech, Bergisch Gladbach, Germany) following the manufacturer’s protocol. The purified EC were seeded in a 0.2% gelatin solution (2% gelatin solution [Sigma-Aldrich, St. Louis, MO, USA] in phosphate-buffered saline [PBS, Merck, Darmstadt, Germany]) in an EC growth medium (EGM II, Lonza, Basel, Switzerland) for further cultivation.

#### 4.1.3. Acetylated-Low-Density Lipoprotein (Ac-LDL) Uptake Assay

6 × 10^4^ EC/cm^2^ were seeded in a gelatin-coated well plate. The medium was supplemented with Ac-LDL (10 µL/mL, Thermo Fisher Scientific, Waltham, MA, USA) and Nuc-Blue (Thermo Fisher Scientific, Waltham, MA, USA) for nuclear labeling. The EC were analyzed using an inverted fluorescence microscope (Leica DMi 8, Leica Microsystems, Wetzlar, Germany).

#### 4.1.4. AGAL Enzyme Activity Assay

A commercially available kit quantified the AGAL enzyme activity in the cultivated EC (Abcam, Cambridge, UK). The samples were treated with 4-methylumbelliferone substrate. Cleavage of this substrate using AGAL causes a fluorescent signal, which was measured using a TECAN Infinite M200 Pro multi-plate reader (Tecan, Männedorf, Switzerland). After normalizing the fluorescent signal to the protein concentration of the sample with a bicinchoninic acid assay (Interchim, Montlucon, France), the sample specific AGAL activity was determined in µU/mg.

#### 4.1.5. EC Tube Formation Assay

8 × 10^5^ EC/cm^2^ were seeded on a pure Cultrex- (R&D Systems Inc., Minneapolis, MN, USA) coated µ-Slide Angiogenesis (Ibidi, Gräfelfing, Germany) in EGM II or EGM II supplemented with 10 µM of SB or 1.32 µL/mL of agalsidase-α (Takeda Pharmaceutical, Tokyo, Japan), respectively. After 3 h of incubation, live cell imaging was started, and photomicrographs were taken every 6 h for up to 48 h. The experiment was conducted using an inverted fluorescence microscope (Leica DMi 8, Leica Microsystems, Wetzlar, Germany) with heating and gas incubation systems (both Ibidi, Gräfelfing, Germany) to ensure cell viability during the experiments. Data analysis was performed using FastTrackAI Angiogenesis (Ibidi, Gräfelfing, Germany).

#### 4.1.6. EC Cultivation under Hypoxic Conditions

EC were seeded two days before the incubation experiment and transferred into a hypoxia chamber made of acrylic glass (30 × 20 × 16 cm) modified according to [22]. The chamber was sealed with gas-tight adhesive tape (DIOP, Rosbach, Germany), and flooded with a hypoxic gas mixture (93% [*v*/*v*] N_2_, 5% [*v*/*v*] CO_2_, and 2% [*v*/*v*] O_2_). The chamber was transferred into a cell culture incubator (Thermo Fisher Scientific, Waltham, MA, USA) and the EC remained in the chamber for 48 h, with a control group cultivated within the incubator outside the hypoxia chamber.

#### 4.1.7. Immunolabeling

The human EC were fixed with 4% paraformaldehyde for 15 min at 37 °C (PFA, Sigma Aldrich, St. Louis, MO, USA) and blocked with 10% fetal calf serum (FCS)/0.1% saponin (Sigma Aldrich, St. Louis, MO, USA) for extranuclear targets and 10% FCS/0.3% Triton X (Sigma Aldrich, St. Louis, MO, USA) for intranuclear targets for 30 min at RT. The primary antibodies were diluted in the respective blocking solution and incubated overnight at 4 °C. The secondary antibodies were diluted in PBS and incubated for 30 min at RT. The human primary and respective secondary antibodies are listed in Appendix A. The immunolabeling procedure was followed by washing with PBS and incubation with 4′,6-diamidino-2-phenylindole (DAPI; 1:10 000, Sigma Aldrich, St. Louis, MS, USA). Labeled EC were mounted with Aqua-Poly/Mount (Polysciences, Warrington, PA, USA) and stored in the dark at 4 °C until further processing.

#### 4.1.8. Expansion Microscopy

Before expansion, the labeled EC were initially imaged. Subsequently, they were incubated with a 0.25% glutaraldehyde solution (SERVA, Heidelberg, Germany) in PBS for 10 min at RT. Gelation was then carried out using a monomer solution comprising 8.635% sodium acrylate, 2.5% acrylamide, 0.15% N,N’-methylenbisacrylamide, 2 M NaCl in PBS, along with 0.2% ammonium persulfate and 0.2% tetramethylethylenediamine (all Sigma Aldrich in St. Louis, MO, USA). The gels were incubated in digestion buffer (50 mM Tris [SERVA, Heidelberg, Germany], 25 mM ethylenediaminetetraacetic acid [Thermo Fisher Scientific, Waltham, MA, USA], 0.5% Triton X [Sigma Aldrich, St. Louis, MO, USA], and 4 U/mL proteinase K [NEB, Ipswich, MA, USA]) for 45 min at 37 °C and washed in PBS, with DAPI included in the second of the three washing steps. Gel expansion was performed in double-distilled water, and the expanded gels were transferred into poly-D-lysine-coated (Sigma Aldrich, St. Louis, MO, USA) chambers (Thermo Fischer Scientific, Waltham, MA, USA). The expansion factor was determined using DAPI and the CD144 signal of the same region and manually aligned using Inkscape V 0.92 (https://inkscape.org, access and download: 14 January 2020). Imaging was performed with an inverted epifluorescence microscope (Leica, DMi8, with Thunder Imaging Software version 3.7.5.24914).

#### 4.1.9. Mitochondrial Morphology

For the assessment of the mitochondrial morphology using immunolabeling against translocase of outer mitochondrial membrane 20 (TOMM20), photomicrographs were acquired using an inverted epifluorescence microscope (Leica, DMi8, with Thunder Imaging Software version 3.7.5.24914). The data were analyzed for selected mitochondrial parameters such as form factor, solidity, and weighted distance using MitoHacker [16]. The “form factor” reflects the shape and branching of the mitochondria, with a lower number indicating round, uniform mitochondria and no elongated mitochondria or intense network structures. The “solidity” describes the degree of sparseness of the mitochondria, which indicates how much of a putative ellipsoid drawn around the mitochondrion is filled by the mitochondrion itself. The “weighted distance” depicts the distance of the center of the mitochondrion to the center of its respective nucleus weighted by the size of the mitochondrial area.

#### 4.1.10. Mitochondrial Membrane Potential

To measure the mitochondrial membrane potential, a JC-10 assay (AAT Bioquest^®^Inc, Sunnyvale, CA, USA) was performed. The EC were cultivated on 24 well plates for five days after MACS and then incubated with 20 µM of JC-10 diluted in 200 μL 1:1 of 4-(2-hydroxyethyl)-1-piperazineethanesulfonic acid buffer (151 mM NaCl, 1.25 mM NaH_2_PO_4_, 2.5 mM KCl, 2 mM CaCl_2_, 1 mM MgCl_2_, 10 mM glucose, pH 7.4), and Hanks’ balanced salt solution (Thermo Fisher Scientific, Waltham, MA, USA) for 30 min at 37 °C in the dark. The fluorescence was measured at 490/525 nm and 540/590 nm using a TECAN Infinite M200 Pro multi-plate reader.

#### 4.1.11. EC Gene Expression Analysis

For the total RNA extraction from the human EC cultures, the miRNeasy mini kit (Qiagen, Hilden, Germany) was used. The RNA was reverse-transcribed using TaqMan^®^ reverse transcription reagents (Applied Biosystems, Darmstadt, Germany). For screening the dysregulated hypoxia- and angiogenesis-associated target genes, we used array plates preloaded with a selection of hypoxia-associated gene assays (TaqMan^®^ array human hypoxia, Applied Biosystems, Darmstadt, Germany). A selection of dysregulated genes was validated using duplex qRT PCR analysis with TaqMan^®^ Fast Advanced Mastermix (Applied Biosystems, Darmstadt, Germany). The human TaqMan^®^ Gene Expression assays were tagged with FAM-MGB reporter dye and the endogenous control tagged with VIC-MGB reporter dye (all Applied Biosystems, Darmstadt, Germany; the assays and respective IDs are listed in Appendix A). qRT PCR analysis was run with the QuantStudio 3 Real-Time PCR System (Applied Biosystems, Darmstadt, Germany) at the following conditions: 2 min, 50 °C; 2 min, 95 °C; (3 s, 95 °C; 30 s, 60 °C) 40×. The performed qRT PCR was acquired using the QuantStudio Design & Analysis software v1.5.1. The relative gene expression was analyzed according to the ΔΔCt method.

### 4.2. Experiments on GLA KO Mouse Model

#### 4.2.1. Mouse Colony

The murine experiments including *GLA* KO mice and WT littermates were approved by the state authority of Bavaria (Regierung von Unterfranken #54/12; #1052/22). The animals were bred and kept at the animal facilities of the Department of Neurology and of the Center for Experimental Molecular Medicine (Zentrum für Experimentelle Molekulare Medizin, ZEMM), University of Würzburg, Germany. Genotype analysis was performed in all mice using the Taq PCR Master Mix Kit (Qiagen, Hilden, Germany) and the following primers: oIMR5947, AGGTCCACAGCAAAGGATTG; oIMR5948, GCAAGTTGCCCTCTGACTTC; and oIMR7415, GCCAGAGGCCACTTGTGTAG (Invitrogen, Carlsbad, CA, USA). Homozygous *GLA* KO and WT mice with identical genetic backgrounds stratified by age (young mice < 6 months, old mice ≥12 months) were investigated. Male and female mice were equally included in all investigated groups after the exclusion of sex differences in our previous study [77].

#### 4.2.2. Tissue Collection

For DRG neuronal cell culture, immunohistochemistry, and gene expression analysis of the murine DRG tissue, whole DRG from the *GLA* KO and WT mouse groups were dissected. The mice were euthanized using deep isoflurane anesthesia (CP-Pharma, Burgdorf, Germany). All available DRG were dissected according to [78] and collected in PBS on ice for neuronal cell culture. L3 and L5 DRG were collected for qRT PCR analysis, while L4 DRG were embedded into an optimal cutting temperature medium (TissueTek^®^, Sakura Finetek, Staufen, Germany) for the immunohistochemistry experiments. The tissue was flash-frozen in liquid-nitrogen-cooled 2-methylbutane (Carl Roth, Karlsruhe, Germany) and stored at −80 °C until further processing. For immunolabeling, the L4 DRG were cut into 10 µm cryosections for cell counting analysis and 50 µm cryosections for blood vessel analysis using a cryostat (Leica Microsystems, Wetzlar, Germany). Three cryosections per mouse were collected per slide for further analysis.

#### 4.2.3. DRG Neuronal Cell Culture

The collected murine DRG were extracted from the dorsal and ventral root nerves and transferred into a DRG medium composed of DMEM/F-12 + GlutaMAX^TM^ (Thermo Fisher Scientific, Waltham, MA, USA), 100 U/mL of penicillin/100 µg/mL streptomycin (PenStrep, Thermo Fisher Scientific, Waltham, MA, USA), and 10% fetal calf serum (FCS, Merck, Darmstadt, Germany). The DRG were incubated in DMEM/F-12 + GlutaMAX^TM^ with 1.92 mg/mL of Liberase TH enzyme (Roche, Basel, Switzerland) at 37 °C and 900 revolutions per min (rpm) for 30 min following incubation with DMEM/F-12 + GlutaMAX^TM^ with 1.92 mg/mL of the Liberase TM enzyme (Roche, Basel, Switzerland) at 37 °C and 900 rpm for 10 min. The digested DRG were resuspended in DMEM/F-12 + GlutaMAX^TM^ and pipetted gently on top of a DMEM/F-12 + GlutaMAX^TM^ and 3.5% bovine serum albumin (BSA, Sigma-Aldrich, St. Louis, MO, USA) mixture building two visible phases. After the centrifugation step at RT and 500 rpm for 10 min, the DRG pellet was resuspended in DRG medium + 10 µg/mL nerve growth factor (NGF, Alomone Labs, Jerusalem, Israel) and pipetted onto poly-D-lysine/laminin-coated coverslips (Corning Inc., Corning, NY, USA). The DRG neuronal cell cultures were kept at 37 °C in an incubator for at least 24 h until further processing.

#### 4.2.4. In Vitro Hypoxia Experiments with DRG Neuronal Cell Cultures

The DRG neuronal cell cultures from the old WT and *GLA* KO mice were placed into the hypoxia chamber, which was sealed with gas-tight adhesive tape (DIOP, Rosbach, Germany). The chamber was flooded with a 2% O_2_ gas mixture (93% [*v*/*v*] N_2_, 5% [*v*/*v*] CO_2_, and 2% [*v*/*v*] O_2_) for 2–3 min. Then, the hypoxia chamber was placed into an incubator at 37 °C for 24 h. As a control condition, the DRG neurons of old WT and *GLA* KO mice were placed outside the chamber within an incubator at 21% O_2_ for 24 h. Afterward, the neurons were either fixed with 4% PFA for the immunohistological analysis or lysed using QIAzol Lysis Reagent (Qiagen, Hilden, Germany) for the gene expression analysis.

#### 4.2.5. Immunolabeling

The murine cryosections were fixed with acetone (Sigma-Aldrich, St. Louis, MO, USA), and the DRG neuronal cell culture was fixed with 4% PFA. Blocking was performed using 10% BSA. The primary antibodies were diluted in 1% BSA/0.3% Triton X (Sigma-Aldrich, St. Louis, MS, USA) for intranuclear targets, and with 1% BSA/0.1% saponin for extranuclear targets overnight at 4 °C. The secondary antibodies were diluted in 1% BSA and incubated for 2 h at RT. The used primary and secondary antibodies are listed in Appendix A. The incubation was followed by washing steps with PBS and incubation with DAPI. The stained sections were mounted with VECTASHIELD^®^ (Vector Labs, Burlingame, CA, USA), and the stained DRG neurons were mounted with Aqua-Poly/Mount. The samples were stored in the dark at 4 °C until further processing.

#### 4.2.6. Gene Expression Analysis

For the total RNA extraction from the murine DRG tissue and DRG neuronal cell culture, the miRNeasy mini kit (Qiagen, Hilden, Germany) was used. The RNA was reverse-transcribed using TaqMan^®^ reverse transcription reagents (Applied Biosystems, Darmstadt, Germany). To screen the dysregulated hypoxia-associated target genes in whole DRG and the DRG neuronal cell cultures, we applied preloaded array plates with a selection of target gene assays (RT^2^ Profiler PCR Array, Mouse Hypoxia Signaling Pathway, Qiagen, Hilden, Germany) to the pooled whole DRG samples from seven old WT and seven old *GLA* KO mice. A selection of the dysregulated genes was validated with the whole DRG tissue and DRG neuronal cell cultures using duplex qRT PCR analysis and TaqMan^®^ Fast Advanced Mastermix (Applied Biosystems, Darmstadt, Germany). The murine TaqMan^®^ Gene Expression assays were tagged with FAM-MGB reporter dye and the endogenous control tagged with VIC-MGB reporter dye (all Applied Biosystems, Darmstadt, Germany; assays and their respective IDs are listed in Appendix A). qRT PCR analysis was run with the 96-well StepOnePlusTM Real-Time PCR System (Applied Biosystems, Darmstadt, Germany) at the following conditions: 2 min, 50 °C; 2 min, 95 °C (3 s, 95 °C; 30 s, 60 °C) 40×. qRT PCR results were acquired using the StepOnePlus software v2.3. Relative gene expression was analyzed according to the ΔΔCt method.

### 4.3. Statistics

Statistical analysis was performed using the SPSS software version 29 (IBM, Ehningen, Germany). Normality was tested using the Shapiro–Wilk test. For a two-group comparison of the normally distributed data, an independent t-test was performed, while for the non-normally distributed data, the Mann–Whitney U test was used. For independent multiple-group comparison and normally distributed data, one-way ANOVA followed by Tukey’s multiple comparison test was used, while the Kruskal–Wallis test followed by Dunn’s multiple comparison test was performed for the non-normally distributed data. For dependent multiple-group comparison, the Friedman test was performed followed by Dunn’s multiple comparison test. The significance was set at a *p*-value of <0.05. Graphical representations were designed using GraphPad PRISM version 9.4.1 (GraphPad Software, Inc., La Jolla, CA, USA). The human data sets are either represented as boxplots visualizing the median value with the upper and lower 75% and 25% quartile, as violin plots, or as points reflecting the median with connecting lines. The murine data are represented as boxplots visualizing the median value with a 95% confidence interval.

## 5. Conclusions

We report hints at increased hypoxia and impaired vascular properties in human FD EC and complementarily in *GLA* KO mouse DRG. Our data support the hypothesis that hypoxia and hypovascularization may be initiators of FD pain. Hypoperfusion caused by the hampered vascularization of the DRG, the neuronal stress caused by imbalances in the mitochondrial morphology and functionality, and ion channel alterations may lead subsequently to pain development in FD (Figure 15).

## Figures and Tables

**Figure 1 ijms-24-15422-f001:**
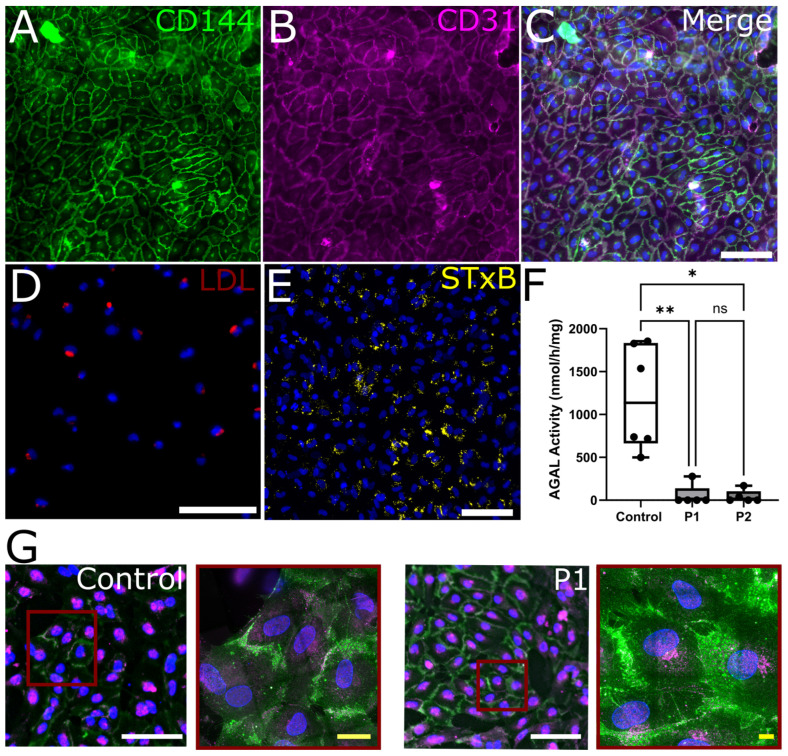
**Characteristics of control-, P1-, and P2-derived EC.** (**A**) Immunolabeling for CD144 and (**B**) CD31, (**C**) both counterstained with DAPI, and (**D**) successful LDL uptake reveal functional EC differentiation. (**E**) Gb3 labeling with STxB in patient EC and (**F**) reduced AGAL activity in P1 and P2 compared to control cells (*n* = 3 per clone, 2 clones per cell line) indicate the distinct Fabry disease specific cellular phenotype. (**G**) Exemplified expansion microscopic images of the EC derived from the healthy control and P1 immunolabeled against CD144 and CD31 show no difference in intermembrane connection between the EC derived from patients with FD compared to EC from a healthy control. Abbreviations: AGAL: α-galactosidase A, CD: cluster of differentiation, DAPI: 4′,6-diamidino-2-phenylindol, LDL: low-density lipoprotein, n.s.: not significant, P1,2: Fabry patients, STxB: Shiga toxin subunit B. Scale bars A-E: 50 µm, G: white 100 µm, yellow 21.5 µm. Statistics: Kruskal–Wallis test in Figure 1F. * *p* < 0.05, ** *p* < 0.01.

**Figure 2 ijms-24-15422-f002:**
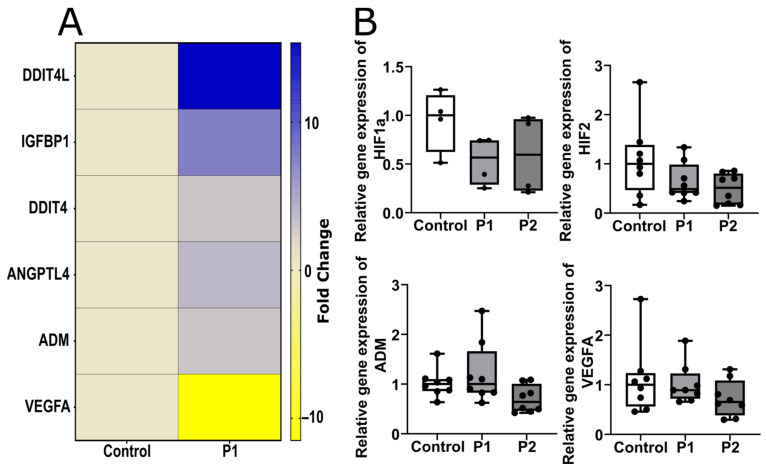
**Hypoxia**–**associated changes in gene expression in control**– **and patient**–**derived EC.** (**A**) Relative expression change in targets associated with hypoxia in pooled EC of control and P1 cells (yellow: downregulation, blue: upregulation). (**B**) Target validation revealed no difference in relative gene expression between control, P1, and P2 (HIF1a: n = 2 per clone, 2 clones per cell line, HIF2, ADM, VEGFA: n = 3 per clone, 2 clones per cell line). Abbreviations: *ADM*: adrenomedullin, *ANGPTL4*: angiopoietin-like 4, *DDIT4*: DNA damage-inducible transcript 4, *DDIT4L*: DNA damage-inducible transcript 4-like, *HIF*: hypoxia-inducible factor, *IGFBP1*: insulin-like growth factor-binding protein 1, P1,2: Fabry patients, *VEGFA*: vascular endothelial growth factor A. Statistics: Kruskal–Wallis test in Figure 2B.

**Figure 3 ijms-24-15422-f003:**
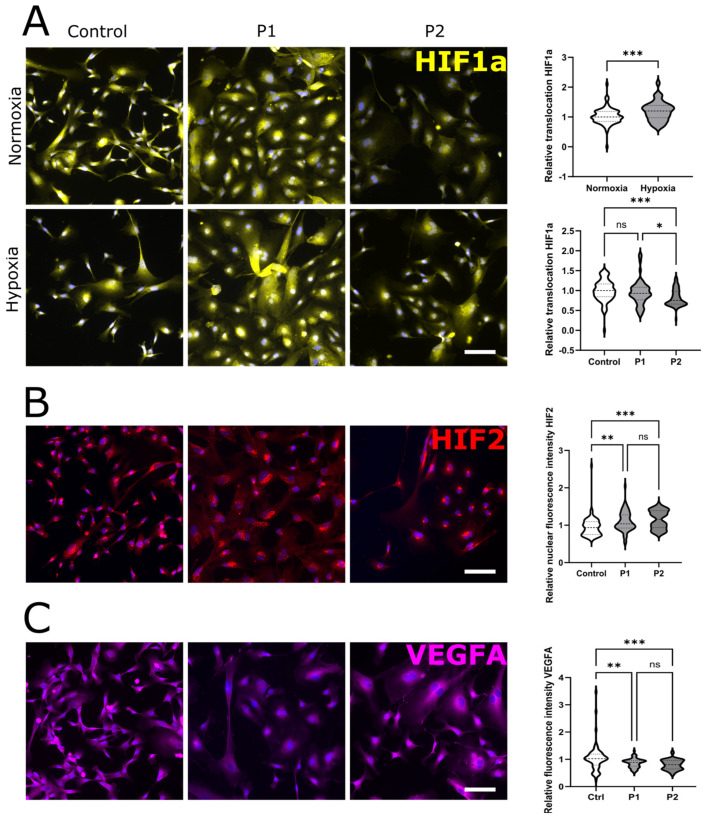
**Changes in protein expression in control and patient EC.** (**A**) Changes in HIF1a expression under hypoxic and normoxic conditions in control, P1, and P2 cells. Hypoxia-associated relative translocation of HIF1a is increased in all cell lines, while patient-specific translocation of HIF1a is reduced in P2. (**B**) Increase in HIF2 signal in P1 and P2 compared to control EC. (**C**) Reduction in VEGFA signal in P1 and P2, compared to the control. (n = 3 per clone, 2 clones per cell line, conducted under normoxic and hypoxic conditions). All photomicrographs were counterstained with DAPI. Abbreviations: DAPI: 4′,6-diamidino-2-phenylindol, EC: endothelial cells, HIF: hypoxia-inducible factor, P1,2: Fabry patients, VEGFA: vascular endothelial growth factor A. Scale bars: 50 µm. Statistics: Mann–Whitney U test in Figure 2A, Kruskal–Wallis test in Figure 2A–C. * *p* < 0.05, ** *p* < 0.01, *** *p* < 0.001.

**Figure 4 ijms-24-15422-f004:**
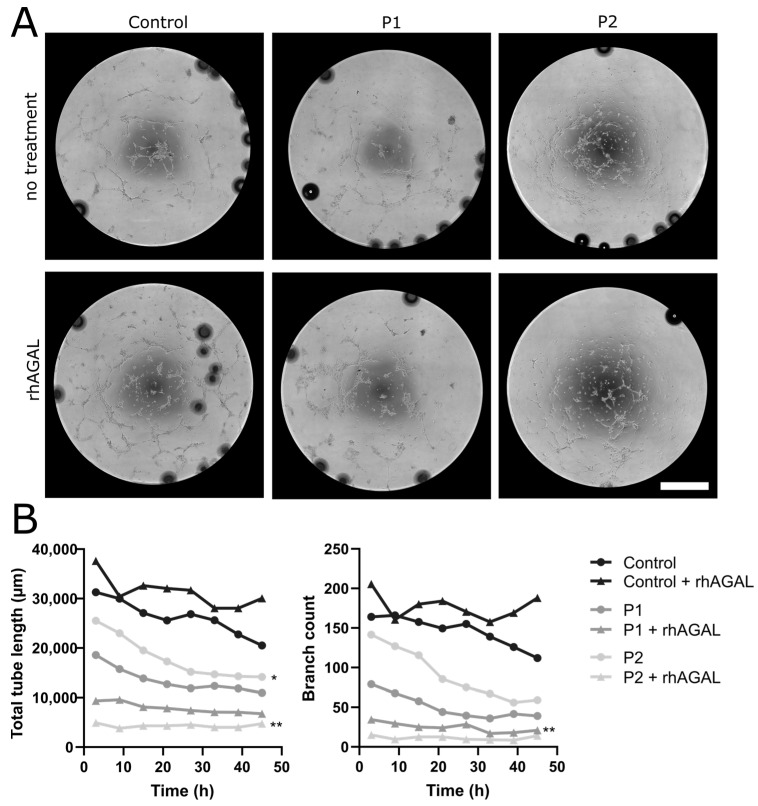
**Changes in tube formation properties in control and patient EC**. (**A**) Reduction in tubular structures in the EC of P1 and P2 compared to the healthy control independent of treatment with agalsidase-α. (**B**) Timepoint analysis showed reduction in total tube length in P2 and branch count in P1 compared to the control EC (n = 3 per clone, 2 clones per cell line, conducted with and without treatment). Abbreviations: rhAGAL: recombinant human agalsidase-α, P1,2: Fabry patients. Scale bar: 1 mm. Statistics: Friedman test. * = *p* < 0.05, ** = *p* < 0.01.

**Figure 5 ijms-24-15422-f005:**
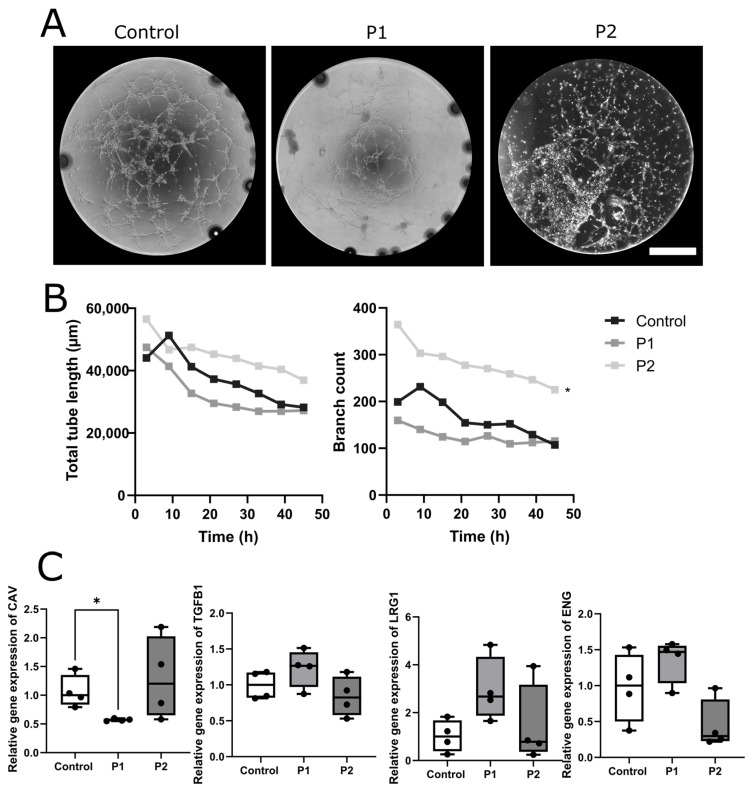
**Recovery of tube formation properties in control and patient EC.** (**A**) Tubular structures after treatment with SB. (**B**) Timepoint analysis of total tube length and branch count was recovered (P1) or even increased (P2) compared to the control EC (n = 3 per clone, 2 clones per cell line, conducted with and without treatment). (**C**) Relative gene expression of the downstream targets of the ALK5-dependent TGFβ pathway (n = 2 per clone, 2 clones per cell line). Abbreviations: *CAV*: caveolin, *ENG*: endoglin, *LRG1*: leucine-rich alpha-2-glycoprotein 1, P1,2: Fabry patients, *TGFβ1*: transforming growth factor β1. Scale bar: 1 mm. Statistics: Friedman test in Figure 5B, Mann–Whitney U test in Figure 5C. * *p* < 0.05.

**Figure 6 ijms-24-15422-f006:**
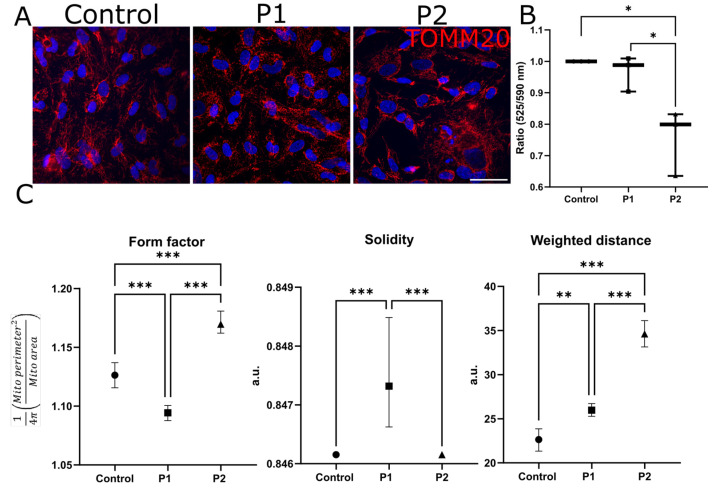
**Mitochondrial functionality of control-, P1-, and P2 derived EC.** (**A**) Immunolabeling for TOMM20, counterstained with DAPI in all three cell lines. (**B**) Mitochondrial membrane potential measured using the ratio of red (525 nm) to green (590 nm) emissions in a JC10 assay, normalized to the control EC, showed a reduced membrane potential in P2 compared to the control and P1’s EC. (**C**) MitoHacker analysis of mitochondrial structure revealed changes in mitochondrial form toward a less complex and branched shape in P1 compared to control EC. P2’s EC showed a more complex and branched shape. Mitochondria in P1’s EC and even more in P2’s EC lay more distant from the nucleus compared to mitochondria in the control EC (n = 3 per clone, 2 clones per cell line). Abbreviations: DAPI: 4′,6-diamidino-2-phenylindol, EC: endothelial cells, P1,2: Fabry patients, TOMM20: translocase of outer mitochondrial membrane 20. Scale bar: 50 µm. Statistics: Kruskal–Wallis test * *p* < 0.05, ** *p* < 0.01, *** *p* < 0.001.

**Figure 7 ijms-24-15422-f007:**
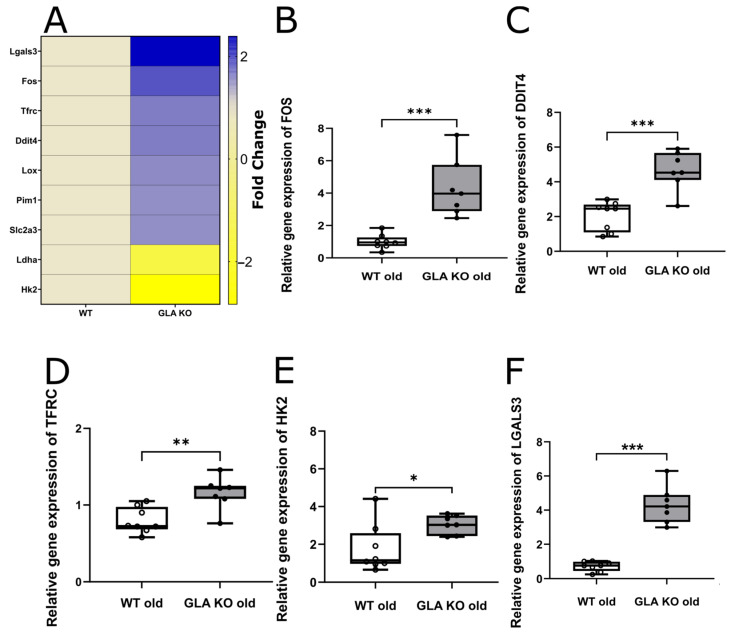
**Hypoxia-associated gene expression analysis in murine DRG of old *GLA* KO and WT mice.** (**A**) Hypoxia-associated gene expression array heatmap displaying a selection of dysregulated hypoxia-associated target genes in DRG tissue of seven pooled old *GLA* KO mice compared to seven pooled old WT mice. Relative gene expression of *DDIT4* (**B**), *FOS* (**C**), *HK2* (**D**), *LGALS3* (**E**), and -*TFRC* (**F**) in the DRG of old WT (○, n = 8 in all graphs) and old *GLA* KO (●, n = 7 in all graphs) mice. Abbreviations: *DDIT4*: DNA damage-inducible transcript 4, FOS: FBJ osteosarcoma oncogene, *GLA* KO: α-galactosidase A knockout, *HK2*: hexokinase 2, *LDHA*: lactate dehydrogenase A, *LGALS3*: galectin 3, *LOX*: lysyl oxidase, *PIM1*: Pim1 proto-oncogene, *SLC2A3*: solute carrier family 2 member 3, *TFRC*: transferrin receptor, WT: wild type. Statistics: independent *t*-test in Figure 7B–D and F. Mann–Whitney U test in Figure 7E. * *p* < 0.05, ** *p* < 0.01, *** *p* < 0.001.

**Figure 8 ijms-24-15422-f008:**
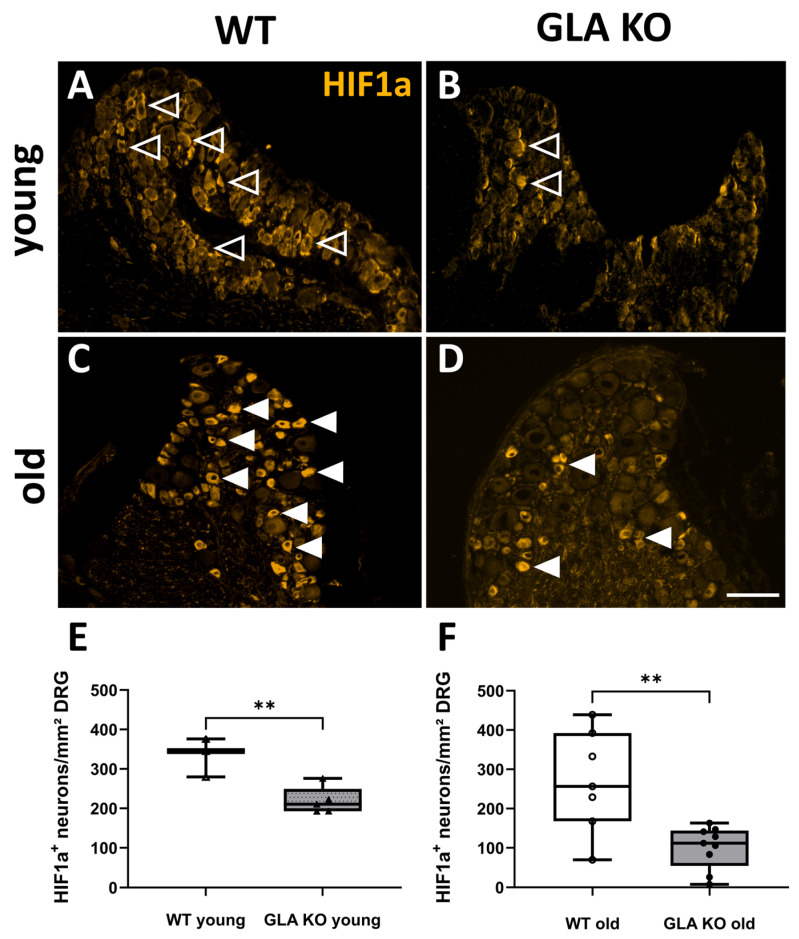
**HIF1a protein distribution in murine DRG neurons**. (**A**–**D**) Representative photomicrographs of HIF1a protein distribution in murine DRG neurons of young and old *GLA* KO and WT mice. HIF1a^+^ DRG neurons in young WT mice ((**A**), empty arrowheads), in young *GLA* KO mice ((**B**), empty arrowheads), in old WT mice ((**C**), full arrowheads), and in old *GLA* KO mice ((**D**), full arrowheads). (**E**,**F**) Quantification of HIF1a^+^ DRG neurons with cytosolic patterns in young WT (△, n = 3), young *GLA* KO (▲, n = 5), old WT (○, n = 7), and old *GLA* KO (● , n = 9) mice. Abbreviations: DRG: dorsal root ganglion, *GLA* KO: α-galactosidase A knock-out, HIF1a: hypoxia-inducible factor 1a, WT: wild type. Scale bar: 100 µm. Statistics: independent *t*-test. ** *p* < 0.01.

**Figure 9 ijms-24-15422-f009:**
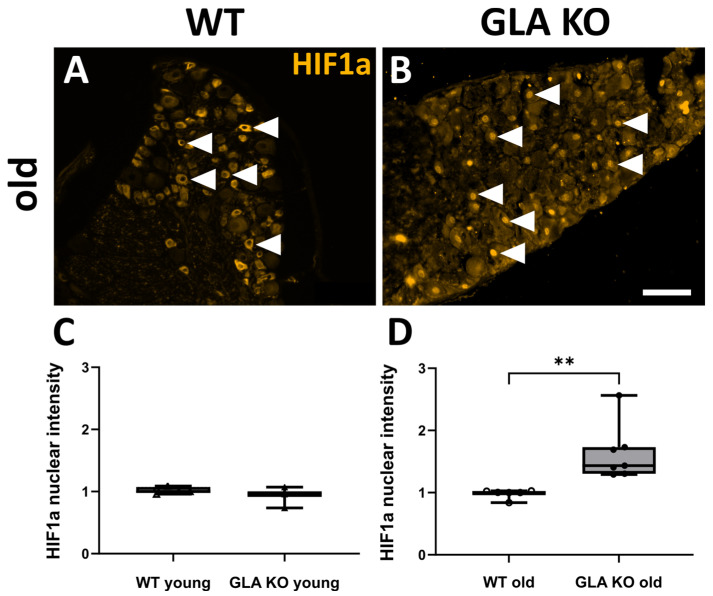
**Nuclear HIF1a intensity measurement in murine DRG neurons.** (**A**,**B**) Representative photomicrographs of cytosolic HIF1a protein distribution in murine DRG neurons of old WT ((**A**), full arrowheads) and nuclear HIF1a protein distribution in murine DRG neurons of old *GLA* KO mice ((**B**), full arrowheads). (**C**,**D**) Nuclear HIF1a intensity measurements in DRG neurons of young WT (△, n = 5), young *GLA* KO (▲, n = 3), old WT (○, n = 6), and old *GLA* KO (●, n = 7) mice. Abbreviations: *GLA* KO: α-galactosidase A knock-out, HIF1a: hypoxia-inducible factor 1a, WT: wild type. Scale bar: 100 µm. Statistics: Mann–Whitney U test. ** *p* < 0.01.

**Figure 10 ijms-24-15422-f010:**
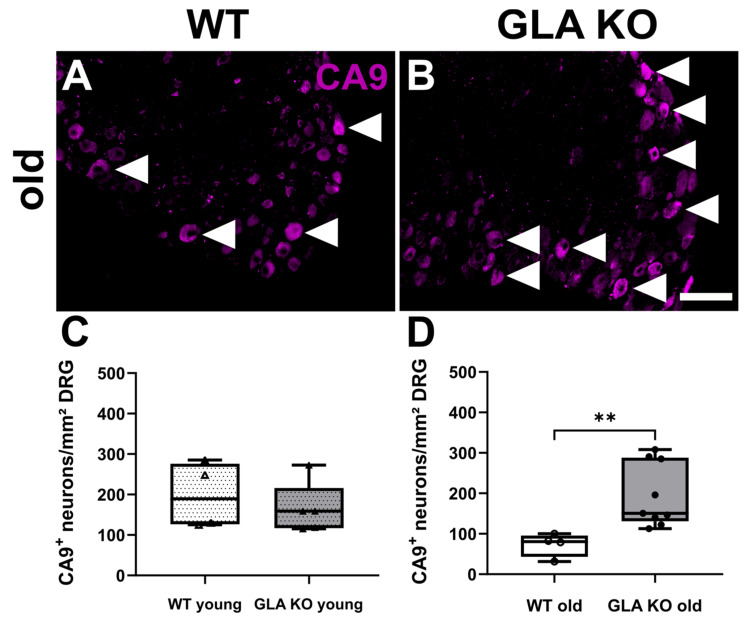
**CA9 protein distribution in murine DRG neurons.** (**A**,**B**) Representative photomicrographs of CA9^+^ DRG neurons in old WT ((**A**), full arrowheads) and old *GLA* KO mice ((**B**), full arrowheads). (**C**,**D**) Quantification of CA9^+^ DRG neurons in young WT (△, n = 4), young *GLA* KO (▲, n = 5), old WT (○, n = 4), and old *GLA* KO (●, n = 9) mice. Abbreviations: CA9: carbonic anhydrase 9, DRG: dorsal root ganglion, *GLA* KO: α-galactosidase A knock-out, WT: wild type. Scale bar: 100 µm. Statistics: Mann–Whitney U test. ** *p* < 0.01.

**Figure 11 ijms-24-15422-f011:**
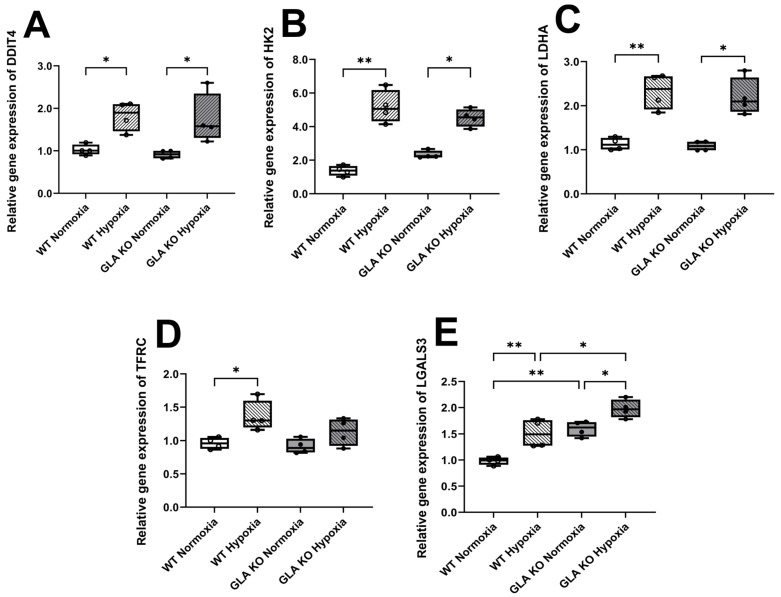
**Hypoxia-associated gene expression analysis in murine DRG neuronal cell cultures of old *GLA* KO and WT mice.** Relative gene expression of *DDIT4* (**A**), *HK2* (**B**), *LDHA* (**C**), *LGALS3* (**D**), and *TFRC* (**E**) in DRG neurons of old WT (○, n = 4 in all graphs) and old *GLA* KO (●, n = 4 in all graphs) mice cultivated for 24 h under normoxic and hypoxic conditions. Abbreviations: *DDIT4*: DNA damage-inducible transcript 4, *GLA* KO: α-galactosidase A knockout, *HK2*: hexokinase 2, *LDHA*: lactate hydrogenase A, *LGALS3*: galectin 3, *TFRC*: transferrin receptor, WT: wild type. Statistics: One-way ANOVA with Tukey’s multiple comparison test in Figure 11A,D,E. Kruskal–Wallis test with Dunn’s multiple comparison test in Figure 11B,C. * *p* < 0.05, ** *p* < 0.01.

**Figure 12 ijms-24-15422-f012:**
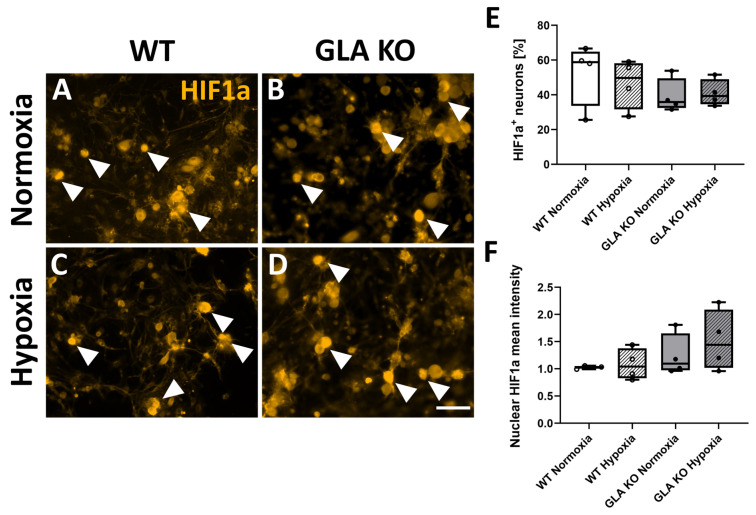
**HIF1a protein distribution in murine DRG neuronal cell culture under normoxic and hypoxic conditions.** (**A**–**D**) Representative photomicrographs of HIF1a^+^ DRG neurons of WT ((**A**,**C**), full arrowheads) and old *GLA* KO mice ((**B**,**D**), full arrowheads) cultivated for 24 h under normoxic and hypoxic conditions. (**E**) Quantification of HIF1a^+^ DRG neurons in old WT (○, n = 4 in all graphs and for all conditions) and old *GLA* KO (●, n = 4 in all graphs and for all conditions) mice under normoxic and hypoxic conditions. (**F**) Mean intensity measurement of nuclear HIF1a of DRG neurons of old WT (○, n = 4 in all graphs and for all conditions) and old *GLA* KO (●, n = 4 in all graphs and for all conditions) mice. Statistical analysis: One-way ANOVA with Tukey’s multiple comparison test. Abbreviations: *GLA* KO: α-galactosidase A knock-out, HIF1a: hypoxia-inducible factor 1a, WT: wild type. Scale bar: 100 µm. Statistics: One-way ANOVA with Tukey’s multiple comparison test.

**Figure 13 ijms-24-15422-f013:**
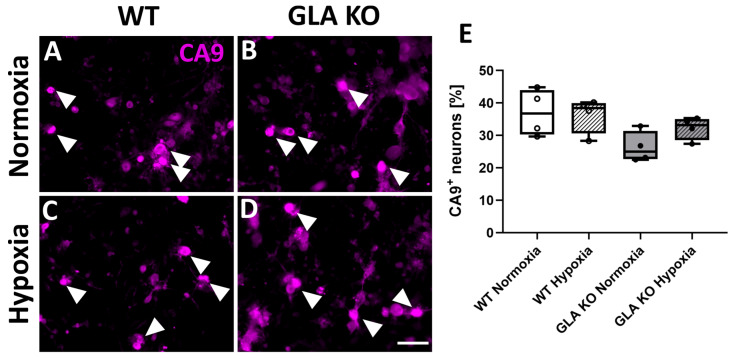
**CA9 protein distribution in murine DRG neuronal cell culture under normoxic and hypoxic conditions.** (**A**–**D**) Representative photomicrographs of CA9^+^ DRG neurons of old WT ((**A**,**C**), full arrowheads) and old *GLA* KO mice ((**B**,**D**), full arrowheads) cultivated for 24 h under normoxic and hypoxic conditions. (**E**) Quantification of CA9+ DRG neurons of old WT (○, n = 4 under all conditions) and old *GLA* KO (●, n = 4 under all conditions) mice under normoxic and hypoxic conditions. Abbreviations: CA9: carbonic anhydrase 9, *GLA* KO: α-galactosidase A knock-out, WT: wild type. Scale bar: 100 µm. Statistics: One-way ANOVA with Tukey’s multiple comparison test.

**Figure 14 ijms-24-15422-f014:**
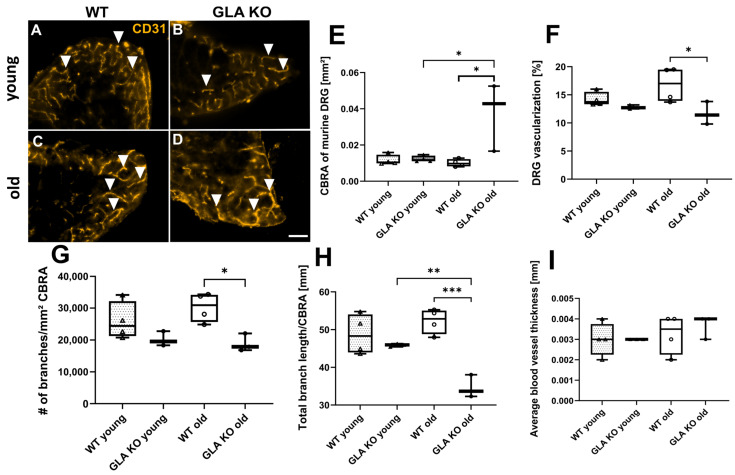
**Vascularization of murine DRG neurons.** (**A**–**D**) Representative photomicrographs of CD31^+^ blood vessels in DRG of young and old WT and *GLA* KO mice (full arrowheads). (**E**) Measurement of DRG CBRA, (**F**) DRG vascularization, (**G**) quantification of blood vessel branches per CBRA, (**H**) measurement of total branch length, and (**I**) measurement of average blood vessel thickness in DRG of young WT (△, n = 4 in all graphs), young *GLA* KO (▲, n = 4 in Figure 14E, n = 3 in all other graphs), old WT (○, n = 4 in all graphs), and old *GLA* KO (●, n = 3 in all graphs) mice. Statistical analysis: Kruskal–Wallis test with Dunn’s multiple comparison test in Figure 14E. One-way ANOVA with Tukey’s multiple comparison test in Figure 14F–I. Abbreviations: CBRA: cell-body-rich area, DRG: dorsal root ganglia, *GLA* KO: α-galactosidase A knock-out, WT: wild type. Scale bar: 100 µm. Statistics: Kruskal–Wallis test with Dunn’s multiple comparison test in Figure 14E. One-way ANOVA with Tukey’s multiple comparison test in Figure 14F–I. * *p* < 0.05, ** *p* < 0.01, *** *p* < 0.001.

**Figure 15 ijms-24-15422-f015:**
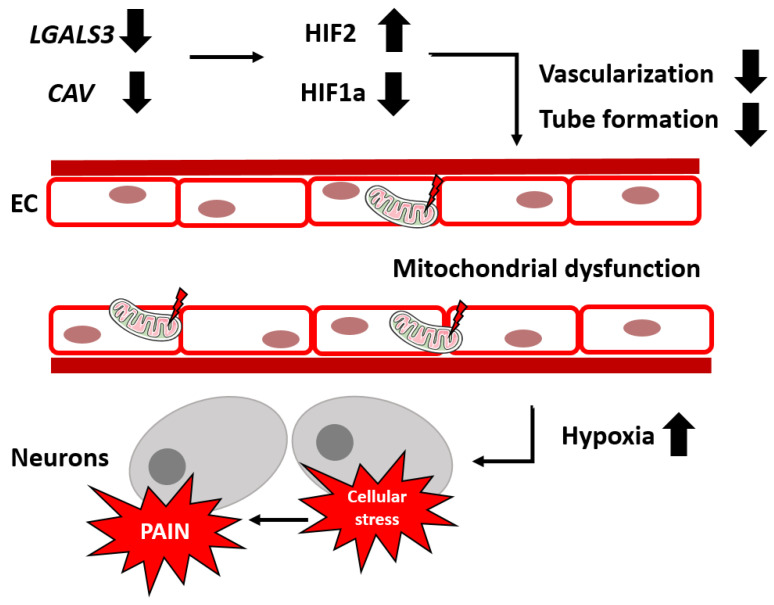
**Impaired vascularization, mitochondriopathy, and hypoxia as potential contributors to FD pain development**. Dysregulation of hypoxia- and angiogenesis-associated target genes, like downregulation of *CAV and LGALS3*, lead to the initiation of hypoxic mechanisms at the protein level including HIF1a and HIF2. These mechanisms might contribute to the reduction of vascularization and tube formation under FD conditions. In combination with mitochondriopathy in FD EC, an increased hypoxic environment for DRG neurons leads further to cellular stress via genetic and protein disturbances. Subsequently, via potential ion channel alteration, FD pain might be induced. Abbreviations: *CAV*: caveolin-1, DRG: dorsal root ganglion, EC: endothelial cells, FD: Fabry disease, HIF1a/2: hypoxia-inducible factor 1a/2, *LGALS3*: galectin 3. Figure 15 contains graphics from https://smart.servier.com/ (last access: 6 August 2023) used under Creative Commons Attribution 3.0 Unported License.

**Table 1 ijms-24-15422-t001:** Subject characteristics.

	P1	P2	Ctrl
**Age**	28	18	59
**Sex**	M	M	M
**Genotype**	c.1069C > T//p.Q357X	c.568delG//p.A190Pfs*2	NA
**FD-associated pain character**	Attacks	No pain	NA
**Cardiomyopathy**	Yes	No	NA
**Nephropathy**	No	No	NA
**FD-specific treatment**	Agalsidase-β	None	NA
**AGAL activity**	0.5 (ref.: 3.4–13.0 nmol/h/mL)	0.04 (ref.: 0.4–1.0 nmol/min/mg/protein)	NA
**Lyso-Gb3**	57.7 (ref.: <0.9 ng/mL)	241 (ref.: <20.1 ng/mL)	NA

**Abbreviations:** A: Alanine, AGAL: α-galactosidase A, C: cytosine, c.: coding, Ctrl: healthy control, FD: Fabry disease, fs: frameshift, G: guanine, lyso-Gb3: lyso-globotriaosylceramide, M: male, NA: not applicable, P: proline, P1,2: Fabry patients, Q: glutamine, ref: reference, T: thymine, X: stop codon.

## Data Availability

The data presented in this study are included in the article. For further information, data sets are available on request from the corresponding author.

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
