# Peer review of "Endothelial Cell Dysfunction and Hypoxia as Potential Mediators of Pain in Fabry Disease: A Human-Murine Translational Approach"

_ijms, 2023, doi:10.3390/ijms242015422_

Round 1

Reviewer 1 Report

In the manuscript “Endothelial cell dysfunction and hypoxia as potential mediators of pain in Fabry disease: a human-murine translation approach”, Klug and co-workers report hints at the role of hypoxia and hypo-vascularization in the Fabry-associated pain development.

Overall, the manuscript is of interest to the field and well organized.

I have some questions to be addressed and some suggestions, reported below.

The name of the gene should always be in italics.

Also, please check that GLA is the gene while AGAL is the enzyme. The authors should double-check the manuscript.

The Materials and Methods are perfectly and detailed.

Results

3.1.3
Why did the authors only perform the comparison in Fig. 2A using P1 and not P2?

How do the authors explain the incongruency between the results of Fig. 2A and 2B?

3.1.5

According to the paragraph title, the authors experimented with normal or hypoxia conditions. In the figures and text, the difference between the conditions is unclear. Also, while the Figure 4 caption states that there is a difference between P1 and P2 and the control, they come to different conclusions in the text (lanes 368-372). Why?

Statistical analysis is missing in Fig. 4B and 5B.

3.1.8 Figure 6B is not cited in the text.

In Figure 7, why is HK2 hypoexpressed in GLA KO in panel A and hyperxpressed in panel E?

Discussion

Recent papers focused on the differential expression in FD, and some of the described results align with those reported by Klug and co-workers. For example, in 10.1186/s12967-023-04475-y, the authors underlined the mitochondrial dysfunction in the GLA KO Zebrafish model, while in 10.3390/ijms23095105, the authors highlighted a differential expression of CAV1 upon treatment of FD fibroblasts. The authors should discuss and cite these and other literature findings, strengthening their paper.

Also, the authors could discuss findings not in line with what they report, such as in 10.3390/jcm9051325, which includes VEGFA in the proteomic signature related to FD but in upregulation.

Author Response

Reply to revision process of the International Journal of Molecular Sciences manuscript “Endothelial cell dysfunction and hypoxia as potential mediators of pain in Fabry disease: a human-murine translation approach”

Reviewer 1:

In the manuscript “Endothelial cell dysfunction and hypoxia as potential mediators of pain in Fabry disease: a human-murine translation approach”, Klug and co-workers report hints at the role of hypoxia and hypo-vascularization in the Fabry-associated pain development.

Overall, the manuscript is of interest to the field and well organized.

I have some questions to be addressed and some suggestions, reported below.

The name of the gene should always be in italics.

Also, please check that GLA is the gene while AGAL is the enzyme. The authors should double-check the manuscript.

We have edited gene names into italics throughout the manuscript. Also, spelling of GLA gene and AGAL enzyme was edited accordingly.

The Materials and Methods are perfectly and detailed.

Results

3.1.3 Why did the authors only perform the comparison in Fig. 2A using P1 and not P2?

We used array analysis to gain an overview of potential differences between control cells and the most severely affected patient, i.e. P1. Genes of interest were then further investigated in both patient cell lines and using more replicates.

How do the authors explain the incongruency between the results of Fig. 2A and 2B?

In Figure 2, panel A, gene expression analysis was first performed using a gene expression array, which was done using pooled EC from P1 and the healthy control. We cannot rule out potential outliers from single samples in this case. Therefore, in panel B, we performed additional single gene expression analysis using qRT-PCR to validate gene expression dysregulation found in the array experiments. Here, single samples were investigated for both patients and the healthy control for two clones each with two to three replicates. These results reflect more reliable differential gene expression in single samples.

3.1.5 According to the paragraph title, the authors experimented with normal or hypoxia conditions. In the figures and text, the difference between the conditions is unclear. Also, while the Figure 4 caption states that there is a difference between P1 and P2 and the control, they come to different conclusions in the text (lanes 368-372). Why?

We specified hypoxia conditions applied in the Methods section 4.1.6 of the original manuscript: We used a hypoxic gas mixture consisting of (93% [v/v] N2, 5% [v/v] CO2, and 2% [v/v] O2). Cells cultivated under normal conditions were in a cell culture incubator with 5% [v/v] CO2. We thank the Reviewer for pointing out the discrepancy between the text and Figure 4 caption, which we have edited accordingly. There is a difference between P1 and the control in the branch count and a difference between P2 and the control in the total tube length. This is independent of treatment.

Statistical analysis is missing in Fig. 4B and 5B.

We have added the statistical analysis to both Figures.

3.1.8 Figure 6B is not cited in the text.

We have included the citation in the revised manuscript (please see line 170).

In Figure 7, why is HK2 hypoexpressed in GLA KO in panel A and hyperxpressed in panel E?

In Figure 7, panel A, gene expression analysis was first performed using a gene expression array, which was performed using pooled murine DRG tissue from old WT and GLA KO mice. We cannot rule out potential outliers from single samples in this case. Therefore, in panel B-E (especially E for HK2 gene expression), we performed additional single gene expression analysis using qRT-PCR to validate gene expression dysregulation seen in the array experiments. Here, single samples were investigated per genotype (whole murine DRG tissue). These results reflect more reliable differential gene expression in single samples. Gambardella et al., 2023 (10.1016/j.isci.2023.106074) reported an increase in protein expression of HK2 in skeletal muscle cells of FD mice and in FD patient-derived fibroblasts supporting the hypothesis that increased HK2 levels might play a role in FD pathophysiology and in our manuscript demonstrated mitochondrial impairment and cellular stress development.

Discussion

Recent papers focused on the differential expression in FD, and some of the described results align with those reported by Klug and co-workers. For example, in 10.1186/s12967-023-04475-y, the authors underlined the mitochondrial dysfunction in the GLA KO Zebrafish model, while in 10.3390/ijms23095105, the authors highlighted a differential expression of CAV1 upon treatment of FD fibroblasts. The authors should discuss and cite these and other literature findings, strengthening their paper.

Also, the authors could discuss findings not in line with what they report, such as in 10.3390/jcm9051325, which includes VEGFA in the proteomic signature related to FD but in upregulation.

We have followed the Reviewer`s suggestion and have included all suggested publications in our revised manuscript together with additional publications as follows:

10.1186/s12967-023-04475-y; Elsaid et al., 2023 included in line 382 with additional changes in the text passage, which adds to the evidence of mitochondriopathy in FD pathophysiology.

10.3390/ijms23095105; Monticelli et al., 2022 included in line 376-379 with additional changes in the text passage, which adds to the effect of ASA on CAV1 protein levels in FD and supports CAV1 as potential molecular target in FD pathophysiology.

10.3390/jcm9051325; Tebani et al., 2020 and additional publications (Ivanova et al., 2023; Sorriento & Iccarino 2021; Zampetti et al., 2013; Do et al.,2020; Lee et al., 2012) included in line 365-369 with additional changes in the text passage, which discuss VEGFA upregulation in several FD models. Further changes in the text passage addresses the contradictory finding in our manuscript regarding reduced VEGFA protein expression.

Reviewer 2 Report

There are a couple of inaccuracies in the manuscript that require attention. Firstly, gene names should consistently be written in italics throughout the text to align with standard scientific writing conventions.

Secondly, there is some inconsistency in the use of "GLA"/"AGAL" to refer to the gene, whereas "GLA" represents the gene, and "AGAL" is the enzyme. Maintaining this distinction consistently throughout the manuscript is essential to ensure clarity among readers regarding the study's terminology and concepts.

Concerning statistics in M&Ms:

1. Is there any reason why the authors used two tests for normality? I would point the attention of the authors to the fact that heterogeneity of variance poses a more significant threat to test validity than normality. In that sense, authors could use Bartlett's test or correct for unequal variance.

2.  I suppose authors used pairwise post-hoc (see e.g. Fig 1 F and FIg3), but I can not find any detail in the Statistics section. Why?

3. Is there any reason the authors used three different visualization paradigms (i.e. violin/box/range plots) when comparing sample estimators?

Results:

Section 3.1.3

The authors should explain why they chose only P1 for the differential expression evaluation.
I am puzzled by the apparent contradiction between Fig. 2A and 2B and how the same is straightforwardly expressed in the main text. How is it possible for the same gene to be first differentially expressed and then not?

Section 3.1.5

The statement in Figure 4 seems to contradicts lines 368-372. Please check.

Figures

In All the figures, the authors should clearly state 1. the sample size and 2. the statistical test they used.

I would  suggest improving the readability of Fig 2A and 7A: the current version has no legend title, making it difficult to reason with the result. Is -10 measured as -logFC? Something else? May I propose to remove the legend altogether and write values inside the rectangles (keeping the color to reinforce the message)?

Author Response

Reply to revision process of the International Journal of Molecular Sciences manuscript “Endothelial cell dysfunction and hypoxia as potential mediators of pain in Fabry disease: a human-murine translation approach”

Reviewer 2:

Comments and Suggestions for Authors

There are a couple of inaccuracies in the manuscript that require attention. Firstly, gene names should consistently be written in italics throughout the text to align with standard scientific writing conventions.

Secondly, there is some inconsistency in the use of „GLA“/“AGAL“ to refer st he gene, whereas „GLA“ represents the gene, and „AGAL“ st he enzyme. Maintaining this distinction consistently throughout the manuscript is essential to ensure clarity among readers regarding the study‘s terminology and concepts.

Please see our reply to the first comment of Reviewer 1 above. We have edited gene names into italics throughout the manuscript. Also, spelling of GLA gene and AGAL enzyme was edited accordingly.

Concerning statistics in M&Ms:

  1. Is there any reason why the authors used two tests for normality? I would point the attention of the authors to the fact that heterogeneity of variance poses a more significant threat to test validity than normality. In that sense, authors could use Bartlett's test or correct for unequal variance.

We only assumed data normality, if found in both, the Kolmogorov-Smirnov and the Shapiro-Wilk test. As the Shapiro-Wilk test is the stricter of both tests, we decided to only state this one in the Methods part (please see line 677). The Bartlett’s test can be used after the confirmation that our data is (approximately) normally distributed, which was rarely the case. We always used the same statistical procedure for standardization which was testing for normality, followed by the decision of the appropriate test.

  1. I suppose authors used pairwise post-hoc (see e.g. Fig 1 F and FIg3), but I can not find any detail in the Statistics section. Why?

We have added the information about used pairwise post-hoc tests according to the multiple comparison analysis described in more detail in section 4.3 Statistics.

  1. Is there any reason the authors used three different visualization paradigms (i.e. violin/box/range plots) when comparing sample estimators?

The different visualization paradigms are due to the variety of samples illustrated. The boxplots are used, if the analysis concludes results from biological and technical replicates of one experiment performed in a row. The violin plots are used to visualize a bigger sample size. In this case, several 100-1000 cells were analyzed plot. Further, violin plots are highly effective in showing the distribution of data points in a data set in a clear and intuitive way. For mitochondrial data, mitochondria in several 100-1000 cells were analyzed per status. Due to outliers in such a big data set, we used the median with a 95% confidence interval to ensure proper visualization of differences.

Results:

Section 3.1.3

The authors should explain why they chose only P1 for the differential expression evaluation.
I am puzzled by the apparent contradiction between Fig. 2A and 2B and how the same is straightforwardly expressed in the main text. How is it possible for the same gene to be first differentially expressed and then not?

Please also see our reply to questions 1 and 2 of Reviewer 1 above.

We used array analysis to gain an overview of potential differences between control cells and the most severely affected patient, i.e. P1. Genes of interest were then further investigated in both patient cell lines and using more replicates.

In Figure 2, panel A, gene expression analysis was first performed using a gene expression array, which was done using pooled EC from P1 and the healthy control. We cannot rule out potential outliers from single samples in this case. Therefore, in panel B, we performed additional single gene expression analysis using qRT-PCR to validate gene expression dysregulation found in the array experiments. Here, single samples were investigated for both patients and the healthy control for two clones each with two to three replicates. These results reflect more reliable differential gene expression in single samples.

Section 3.1.5

The statement in Figure 4 seems to contradicts lines 368-372. Please check.

Please also see our reply to question 3 of Reviewer 1 above.

We thank the Reviewer for pointing out the discrepancy between the text and Figure 4 caption, which we have edited accordingly. There is a difference between P1 and the control in the branch count and a difference between P2 and the control in the total tube length. This is independent of treatment.

Figures

In All the figures, the authors should clearly state 1. the sample size and 2. the statistical test they used.

As recommended, we included in all figures the sample size and used statistical tests.

I would suggest improving the readability of Fig 2A and 7A: the current version has no legend title, making it difficult to reason with the result. Is -10 measured as -logFC? Something else? May I propose to remove the legend altogether and write values inside the rectangles (keeping the color to reinforce the message)?

We have included the legend title to Fig 2A describing fold change.

Round 2

Reviewer 2 Report

 I am satisfied with the changes the authors made in response to my previous feedback. The modifications and clarifications they implemented have  improved the overall quality of the paper, addressing the concerns I raised during the first round of review.